# Multidisciplinary Rehabilitation after Hyaluronic Acid Injections for Elderly with Knee, Hip, Shoulder, and Temporomandibular Joint Osteoarthritis

**DOI:** 10.3390/medicina59112047

**Published:** 2023-11-20

**Authors:** Lorenzo Lippi, Martina Ferrillo, Alessio Turco, Arianna Folli, Stefano Moalli, Fjorelo Refati, Luca Perrero, Antonio Ammendolia, Alessandro de Sire, Marco Invernizzi

**Affiliations:** 1Department of Health Sciences, University of Eastern Piedmont “A. Avogadro”, 28100 Novara, Italy; lorenzolippi.mt@gmail.com (L.L.); alessio.turco.phys@gmail.com (A.T.); arianna.folli23@gmail.com (A.F.); stefano.moalli@libero.it (S.M.); 20032817@studenti.uniupo.it (F.R.); marco.invernizzi@med.uniupo.it (M.I.); 2Translational Medicine, Dipartimento Attività Integrate Ricerca e Innovazione (DAIRI), Azienda Ospedaliera SS. Antonio e Biagio e Cesare Arrigo, 15121 Alessandria, Italy; 3Department of Health Sciences, University of Catanzaro “Magna Graecia”, 88100 Catanzaro, Italy; 4Neurorehabilitation Unit, Azienda Ospedaliera SS Antonio e Biagio e Cesare Arrigo, 15121 Alessandria, Italy; lperrero@ospedale.al.it; 5Department of Medical and Surgical Sciences, University of Catanzaro “Magna Graecia”, 88100 Catanzaro, Italy; ammendolia@unicz.it (A.A.); alessandro.desire@unicz.it (A.d.S.); 6Research Center on Musculoskeletal Health, MusculoSkeletalHealth@UMG, University of Catanzaro “Magna Graecia”, 88100 Catanzaro, Italy

**Keywords:** osteoarthritis, pain, joint pain, musculoskeletal rehabilitation, viscosupplementation, prehabilitation, hyaluronic acid, rehabilitation

## Abstract

Osteoarthritis (OA) is a prevalent degenerative joint condition characterized by cartilage deterioration, joint inflammation, and functional limitations, particularly impacting the elderly population. Rehabilitation and hyaluronic acid (HA) injections are common therapeutic approaches routinely used in clinical practice, but their synergistic potential is far from being fully characterized. Thus, the aim of this narrative review was to elucidate the multilevel benefits and synergies of integrating these two approaches in multidisciplinary OA rehabilitation. This narrative review follows the scale for the assessment of narrative review articles (SANRA) criteria and involves a comprehensive literature search from July to August 2023. Two independent reviewers screened studies, including those involving human subjects with OA, rehabilitation strategies, and outcomes following HA injection, published in English. Results: HA injections might improve joint biomechanics, reducing friction, absorbing shocks, and potentially regulating inflammation. Rehabilitation plays a pivotal role in strengthening muscles, increasing the range of motion, and enhancing overall function. Optimizing rehabilitation following HA injection might provide additional benefits in joint health. OA management requires a multidisciplinary approach integrating HA injections, rehabilitation, and personalized care. Challenges in patient adherence and healthcare resources currently exist, but emerging technologies offer opportunities to enhance patient engagement and monitoring optimizing sustainability and outcomes of patients with knee, hip, shoulder, and temporomandibular joint OA.

## 1. Introduction

OA is a degenerative joint disease that affects millions of people worldwide, with an increasing prevalence due to the aging population [1]. It is characterized by the progressive degradation of articular cartilage, joint inflammation, and changes in the surrounding structures, leading to pain, stiffness, and functional limitations. Considering its chronic nature and its impact on activities of daily living (ADL), OA is currently considered a significant burden on both patients and healthcare systems [2,3].

Patients with OA might be characterized by a multicomponent disability, affecting their independence to perform activities such as walking, climbing stairs, and even basic self-care tasks. These limitations in mobility and physical function have a crucial impact on patients’ quality of life, leading to reduced social participation and increased dependence on assistance [4,5].

In this context, an early diagnosis and a multidimensional approach are necessary for improving outcomes in OA patients [6,7].

Though surgical interventions are often considered for end-stage disease, non-surgical strategies play a crucial role in early-to-moderate stages of OA [8]. In this scenario, several pharmacological strategies may be used, including corticosteroid (CS), platelet-rich-plasma injections, and hyaluronic acid (HA) [9].

More in detail, HA has been shown to have different properties, including the modulation of cellular functions, suppression of pro-inflammatory mediators, and attenuation of the nociceptive response in arthritic joints [10]. Interestingly, a recent systematic review and meta-analysis by He et al. [11] assessed the different therapeutical effects of CS injections compared to HA injections. Interestingly, CS is more effective in the short term (up to 1 month), whereas intriguing implications of HA intra-articular injection were underlined in the long term (up to 6 months). Thus, different combined therapies have been proposed in the literature, showing superior pain-relieving effects of combined HA and CS injections in both the short term and long term [12].

Concerning HA intra-articular injection, it has gained recognition as a treatment option aimed at inducing pain relief, improving joint function, and potentially slowing disease progression [13]. HA is a naturally occurring polysaccharide that is abundantly found in various tissues. In the synovial fluid, it plays a crucial role in joint health by providing lubrication, shock absorption, and maintaining the structural integrity of the extracellular matrix [14]. HA possesses unique characteristics that make it well-suited for improving the rehabilitation management of OA. Its high viscosity allows it to reduce friction between joint surfaces and facilitate smooth movement. Moreover, HA has hydrophilic properties, enabling it to absorb water and maintain joint hydration, which is vital for optimal joint function [15]. Several HA molecules are currently available in the clinical management of OA, including linear and cross-linked HA. It should be noted that HA naturally occurring in healthy synovial fluid is linear, whereas cross-linked HA allows achieving higher molecular weight compounds [16]. Among the available HA formulations, the current literature describes different biological effects which are related to HA molecular weight, with molecules less than 5 kDa having a potential pro-inflammatory effect [17,18] and compounds having a molecular weight more than 800 kDa enhancing a pro-resolving effect [18,19]. In more detail, the pro-inflammatory effect of low-molecular-weight HA would be driven by an up-regulatory effect of pro-inflammatory genes, including NOS2, TNF, IL12B, and CD80, increasing the secretion of nitric oxide and TNF-α [17]. Conversely, the anti-inflammatory effect of high-molecular-weight HA would be driven by the up-regulation of pro-resolving genes, including arg1, IL10, and MRC1 [17].

In addition, a strong body of evidence describes the comprehensive therapeutic effect of high-molecular-weight HA, targeting both pain and function [20,21]. Apparently, these outcomes are targeted not only via the intrinsic HA physical properties, as the benefits of high molecular weight HA exceed the half-life of the compounds [19].

These characteristics of HA might enhance rehabilitation processes by improving the biomechanical proprieties of joint structures and reducing patients’ individual barriers to joint movement [22].

On the other hand, rehabilitation is considered as a cornerstone of the non-surgical management of OA patients. By targeting specific impairments and functional deficits associated with OA, rehabilitation strategies aim to improve joint stability, strengthen muscles around the affected joint, increase the range of motion, and enhance overall physical function [23].

Despite these considerations, to date, there is still a large gap of knowledge about the potential rehabilitation strategies enhancing the benefits of HA intra-articular injection, and the optimal timing, frequency, duration, and intensity are far from being fully characterized.

Therefore, the aim of this narrative review was to highlight the role of rehabilitation strategies following HA intra-articular injection to provide a broad overview of the evidence supporting the effects of a multidisciplinary and tailored approach in terms of pain relief, joint function, and the quality of life of patients with knee, hip, shoulder, and temporomandibular joint OA.

## 2. Materials and Methods

This narrative review followed the scale for the quality assessment of narrative review articles (SANRA) criteria [24], and the SANRA score of this narrative review is reported in Appendix A. A comprehensive literature search was conducted using databases such as PubMed/MEDLINE, Web of Science (WoS), and Scopus. The search strategy included relevant MESH terms, including “Osteoarthritis,” “Hyaluronic Acid Injection”, “Hyaluronic Acid”, “Intra-Articular Injection”, “Viscosupplementation”, “Rehabilitation,” Physical Therapy”, “Physical and Rehabilitation Medicine”, Physical Exercise”, and “Physical Activity” (Table 1).

More in detail, the search strategy for each single database is reported in detail in Table 2.

Two independent reviewers conducted the screening process (MF and AT), and any disagreements were resolved through consultation with a third reviewer (LL) to reach a consensus.

The literature search was performed from July 2023 to August 2023, including papers that were published within the last three decades and that addressed the following research question: “The role of rehabilitation following HA intra-articular injection for OA”. Specifically, the eligibility criteria only included studies involving human subjects with OA, studies addressing the rehabilitation strategies and outcomes following HA injection, and studies published in the English language. Exclusion criteria included studies in languages other than English; studies without full-text availability; studies involving animal subjects; and conference abstracts, medical degrees, master, or doctoral theses.

Data extraction and synthesis were conducted using a qualitative method. The reviewers (MF and AT) independently extracted relevant information from the included studies and synthesized the findings. In case of disagreement, a third reviewer (LL) was consulted to ensure accuracy and consensus. Given the heterogeneity of the included studies and the narrative review design, a qualitative synthesis approach was used, and the findings were presented in a narrative manner.

## 3. HA-Induced Biochemical Changes in Osteoarthritis Joints

OA is a highly prevalent and disabling musculoskeletal disorder affecting the physical functioning and health-related quality of life (HR-QoL) of millions of people worldwide [1,4,5]. In this context, a deep understanding of the OA biological processes underlying disease progression is needed in the management of the disease. The crucial target for therapeutic intervention in OA is the intricate biochemical environment within the joint space, which governs the viscoelastic properties crucial for joint function and health [25]. The viscoelastic properties, encompassing viscosity and elasticity, directly influence the joint’s capacity to absorb shock, distribute mechanical loads, and maintain smooth articulation [26]. Perturbations in these properties contribute significantly to the degenerative changes characterizing OA [27].

Interestingly, HA is a fundamental constituent of synovial fluid that is gaining substantial attention for its potential to induce biochemical modifications impacting the viscoelastic properties of the joint [28]. HA’s ability to regulate synovial fluid viscosity and modulate chondrocyte metabolism is crucial for joint homeostasis [29]. It has been suggested that HA’s viscoelastic properties might effectively absorb and dissipate mechanical shocks, reducing the impact on the articular surfaces [30]. This shock-absorbing capacity contributes to the preservation of joint integrity and minimizes further structural damage to OA joints. In addition, the HA lubricating properties facilitate smooth joint articulation by reducing friction between articular surfaces [14]. This lubrication enhances joint mobility and helps alleviate the mechanical pain associated with OA [31].

Despite HA injections being widely used to restore viscoelastic balance and mitigate cartilage degeneration [32], growing attention is rising toward their anti-inflammatory impact on joint environment. In more detail, HA injections may help to down-regulate the release of pro-inflammatory cytokines, such as interleukins and tumor necrosis factor-alpha (TNF-α), currently considered a key target to reduce inflammatory pain in OA patients [15,33,34]. 

Lastly, HA injections may provide a protective effect on chondrocytes, the cells responsible for maintaining cartilage integrity [35]. By influencing chondrocyte metabolism, HA could support the maintenance and repair of cartilage tissues [29,36]. Concurrently, HA injections could potentially modulate the composition and structure of the cartilage matrix, influencing its biomechanical properties and overall resilience to mechanical load. Moreover, HA has been implicated in promoting the synthesis of proteoglycans, essential components of cartilage that contribute to its structural and functional properties [37]. 

## 4. Role of Physical Exercise for Osteoarthritis

Physical exercise plays a pivotal role in the comprehensive management of OA, contributing to improve musculoskeletal health and overall well-being [38,39]. In particular, engaging in regular exercise helps strengthen the surrounding musculature, providing enhanced joint stability and reducing cross-frictional forces during movement [40]. This mechanism not only promotes smoother joint articulation, but also minimizes the risk of excessive wear and tear on articular surfaces [41]. Furthermore, physical exercise plays a key role in weight management and represents a crucial target for an effective integrated management of OA patients [42,43]. Maintaining a healthy weight is paramount, as it reduces the mechanical load on weight-bearing joints, lessening strain and potential damage [43]. Weight loss achieved through exercise alleviates the burden on afflicted joints, effectively contributing to pain reduction and improved physical functioning [43,44].

In addition, regular physical activity has been associated with the reduction of systemic inflammation biomarkers, such as C-reactive protein (CRP) and interleukin-6 (IL-6), which are intricately linked to OA pathogenesis [45,46]. Moreover, exercise promotes the production of anti-inflammatory cytokines and enhances the secretion of lubricating synovial fluid [47,48], thereby complementing the actions of HA injections.

Considering the specific needs of OA patients, guidelines often recommend low-impact physical activities such as swimming, cycling, and gentle aerobics [49]. These activities minimize joint strain while maximizing cardiovascular fitness and muscular strength. Interestingly, the potential synergisms between HA injections and physical exercise might represent a hot topic in the literature, since HA injections might further reduce forces on joint surfaces, complementing the effects of exercise and offering a comprehensive approach to improving OA joint function [29,36].

In this scenario, the temporomandibular joints (TMJ) OA and osteoarthrosis are also considered as forms of degenerative joint disease [49,50,51]. In more detail, temporomandibular disorders (TMD) are a group of musculoskeletal diseases involving masticatory muscles, TMJ, and/or associated structures [49,50,51]. In this scenario, TMJ degenerative joint disease is also known as TMJ osteoarthritis and is a TMD subtype that may lead to pain, headache, and dysfunction, and may reduce oral-health-related quality of life [52]. The prevalence of arthrogenous TMD in the general population was more than 30% in adults and particularly in the elderly [53]. Among the minimally invasive approaches to improve this condition, intra-articular injections of HA have been investigated [49,50,51].

The dynamic interplay between HA-induced biochemical alterations, exercise-induced anti-inflammatory responses, and the potential synergisms of these interventions holds promising implications for OA management. By addressing multiple facets of OA pathophysiology, a multidisciplinary rehabilitation approach might enhance the tailored management of OA patients by targeting the multicomponent disability characterizing these subjects. In this context, bridging the gap between innovative scientific insights and effective clinical practice is necessary for a sustainable rehabilitation intervention improving not only pain intensity and physical functioning, but also impacting the long-term quality of life of patients with OA.

## 5. Rehabilitation Strategies Following Hyaluronic Acid Intra-Articular Injection

### 5.1. Knee Joint

Knee OA (KOA) is one of the most common locations of the disease, significantly impacting functional ability and quality of life. In the past decades, HA intra-articular injection has emerged as a non-surgical treatment option for KOA, aimed at reducing pain and improving joint function [54].

Several guidelines have provided specific recommendations regarding intra-articular infiltrative procedures. The Osteoarthritis Research Society International (OARSI) [55] emphasizes the potential for long-term pain control (over 12 weeks) through viscosupplementation with HA, offering better safety profiles compared to corticosteroids and no contraindications in frail patients (level 2 evidence). The American Academy of Orthopaedic Surgeons (AAOS) [56] suggested moderate recommendations for the use of HA in knee osteoarthritis, considering its widespread use in clinical practice. Similarly, the European Alliance of Associations for Rheumatology (EULAR) [57] supports the potential efficacy of HA treatment in knee OA.

On the other hand, the benefits of HA injections might be integrated and further optimized by a multidisciplinary rehabilitation approach. In this context, strong evidence supports the effects of tailored exercise programs in improving the functional outcomes of patients with KOA [58,59]. Interestingly, as reported in the systematic review conducted by Tanaka et al. [60], a multicomponent exercise program that includes range of motion exercises, such as knee flexion and extension, helps improve joint mobility and reduce stiffness. By targeting muscles surrounding the knee joint, such as the quadriceps, hamstrings, and calf muscles, stretching routines help alleviate muscle tightness and reduce joint stress [61]. Static stretching, dynamic stretching, and proprioceptive neuromuscular facilitation (PNF) techniques are commonly employed [62]. 

On the other hand, the study by Kılıç et al. [63] examined the impact of an aerobic exercise program combined with physiotherapy on postmenopausal women with knee OA. The results demonstrated significant improvements in pain scores; performance on OA-specific physical performance tests, such as the 40 m Fast-Paced Walk Test, 30 s Chair Stand Test, and Stair Climb Test, as well as outcomes in the six-minute walk test; and scores on the Western Ontario and McMaster Universities Osteoarthritis Index (WOMAC). Similarly, a study by Ettinger et al. [64] investigated the effects of structured exercise programs, including aerobic exercise, on self-reported disability in older adults with knee OA. Both the aerobic exercise group and the resistance exercise group showed significant reductions in disability scores, improvements in pain, and enhanced physical function compared to the health education group. Figure 1 shows the different approaches currently used in knee OA management.

Since low-impact physical activity has been proposed to play a pivotal role in KOA rehabilitation, aquatic exercises have been proposed to improve outcomes in these patients [65]. In more detail, the systematic review with meta-analysis by Song et al. [66] underlined that swimming or water aerobics can improve joint mobility, enhance muscle strength, and alleviate pain. Despite these considerations, water-based exercises might not be feasible in all settings and the need for specific equipment might represent the main barrier to the growth of this approach. 

On the other hand, in recent years, growing attention has risen toward emerging technologies, including exoskeletons or virtual reality systems, offering targeted assistance and feedback during exercise sessions [67]. These exercises help improve joint kinematics, enhance muscle coordination, and provide a motivating environment for patients [68,69].

Although the current literature highlights a gap of knowledge in the field of robotic-assisted rehabilitation combined with a conservative approach, this novelty approach might offer promising results to enhance patient engagement and offer sustainable intervention for a highly prevalent disease such as KOA [70].

Despite these considerations, the timing for rehabilitation after HA injection for knee OA may vary depending on individual factors. Experts generally suggest a brief period of relative rest, typically lasting between 12 to 24 h. During this time, patients are advised to engage in slow walking or cycling, while avoiding high-impact activities [71]. This might allow HA to settle within the joint and reduce the risk of any adverse effects. Moreover, attention should be paid to patients receiving anticoagulant medications or with blood disorders, and low-impact activity should be preferred, reducing stress on the joint in the short period after the injection [71].

Altogether, these findings suggested that a comprehensive treatment approach improves physical function, exercise tolerance, and physical performance in knee OA patients. Everyone’s rehabilitation program should be personalized based on the patient’s specific needs, taking into consideration their functional limitations, disease severity, and treatment goals.

### 5.2. Hip Joint

Hip OA is a common condition that affects the mobility and quality of life of older adults. Although the up-to-date literature describes controversial effects of HA injections in patients with hip OA [55,56,57], recent research has now highlighted promising results of ultrasound-guided procedures [72], in addition to exercise therapy also [73]. 

In this context, physical exercise is a key component of hip OA rehabilitation. Range-of-motion exercises, such as hip flexion and extension, abduction, and internal/external rotation, aim to improve joint mobility and reduce stiffness. Stretching exercises targeting the muscles surrounding the hip joint, such as the hip flexors, adductors, and external rotators, are crucial for improving flexibility and joint range of motion, and should be performed within a pain-free range and tailored to the individual’s specific limitations [50]. In accordance with knee joints, aquatic exercise provides a safe and low-impact environment for hip OA rehabilitation. Water-based exercises, such as water walking, leg kicks, and hip circles, offer resistance for muscle strengthening while reducing joint stress [74].

When HA injections are combined with physical exercise or rehabilitation, the synergistic effects can lead to better outcomes and improved quality of life for individuals with hip OA [29]. Interestingly, in the study by Mauro et al. [73], the authors aimed to evaluate the efficacy of ultrasound image-guided intra-articular injections of HA in addition to exercise therapy as a treatment modality for hip OA. The study demonstrated that ultrasound-guided HA injections combined with exercise therapy led to a substantial and sustained reduction in hip OA-related pain during activity, with significant improvements in hip disability, daily functioning, and decreased NSAIDs intake, offering a promising approach for managing hip OA symptoms. 

Altogether, these findings underline the importance of a multicomponent rehabilitation approach leading to overcoming pain barriers to therapeutic exercise and providing synergic effects in physical functioning and physical performance improvements. In this context, attention should be paid to the sustainability of a multicomponent approach and digital technologies might be considered a promising tool to reduce the expenditure and improve the tailored management of patients with hip OA [75,76]. 

### 5.3. Shoulder Joint

The shoulder OA disabling conditions lead to musculoskeletal pain and impairment in the reaching function of the upper limb [77]. Despite HA injections being explored as a treatment option for shoulder OA, strong evidence supporting this therapeutic intervention is still lacking [78]. Moreover, the AAOS clinical practice guidelines for the management of glenohumeral joint OA provide recommendations against the use of HA injections [79]. As a result, to the best of our knowledge, no previous study assessed the role of a multicomponent rehabilitation intervention including HA injection combined with therapeutic exercise, although rehabilitation strategies play a pivotal role in the non-surgical management of shoulder OA [80]. 

Taken together, there is a clear need for further research to investigate the potential benefits of an integrated approach, incorporating HA injections in therapeutic exercise protocols to enhance the benefits induced by rehabilitation and reduce the functional implications of shoulder OA.

### 5.4. Temporomandibular Joint

HA visco-supplementation was reported to have chondroprotective, anti-inflammatory, and lubricating effects in TMJ diseases also [51]. Indeed, OA TMJ patients were found to have a decreased molecular weight of HA in the synovial fluid, and HA injections have been shown to induce synovial cells to synthesize endogenous acid and to promote the repair of cartilage tissue [81].

Systematic reviews evaluated the effects of HA injections and showed effectiveness in terms of pain relief in short- and long-term follow-up [82].

A randomized controlled trial (RCT) by Gencer et al.[83] evaluated the effects of TMJ injections with three different anti-inflammatory agents: HA, betamethasone, and tenoxicam. The results showed that patients who underwent HA injections had significantly better pain relief compared to the other groups after 6 weeks (*p* < 0.05). Moreover, a systematic review reported no differences between HA injections alone or in combination with arthrocentesis and/or arthroscopy in relieving pain [84].

Also, conservative approaches (e.g., physical therapy, splint therapy) were shown to improve pain and functionality, but there is evidence that minimally invasive procedures, including HA injections, may be more effective than conservative treatments for both pain reduction and the improvement of maximum mouth opening. In this context, a recent RCT compared the efficacy of HA injection combined with home physical exercise to physical exercise alone [85]. The crepitus ratio in the combined treatment group improved significantly compared to the baseline (*p* < 0.05), whereas no significant improvement was observed in the physical treatment group [85].

In conclusion, TMJ injections with HA can represent a possible alternative in the complex management of TMD patients suffering from chronic pain and/or not responding to conservative approaches, considering that the TMJ injection procedure is minimally invasive with a low number of complications. 

## 6. Prehabilitation, Physical Exercise, and Pain Relief

Prehabilitation for OA is an emerging concept recently gaining attention in the optimization of the OA management of older adults [86,87,88]. This novelty multidimensional strategy aims at enhancing patients’ physical and psychological well-being before surgery or medical interventions, integrating several components that collectively contribute to better outcomes and minimizing complications. 

In more detail, the recent review by Konnyu et al. [86] highlighted the impact of prehabilitation before total knee arthroplasty and total hip arthroplasty. Compared to no prehabilitation, prehabilitation shows potential implications in enhancing muscle strength and reducing the length of hospital stays [86]. Although there is insufficient evidence to characterize which patients would benefit most, several rehabilitation interventions have demonstrated comparable improvements in pain, strength, ADL, and quality of life [86,87,88]. The key components of a comprehensive prehabilitation approach include exercise training, nutritional support, pain management, psychological support, education, and counseling.

In this context, engaging patients in tailored aerobic, resistance, and flexibility exercises might enhance musculoskeletal recovery and facilitate postoperative processes. According to a recent meta-analysis [88], significant benefits were reported in terms of the physical functioning and length of stay in patients receiving physical exercise interventions before surgery. On the other hand, exercise intervention is crucial for weight management, which represents a key target for reducing complications following surgical procedures that were highly prevalent in obese patients [89]. In particular, physical exercises associated with nutritional counseling have a role in optimizing the body’s readiness for upcoming interventions because of their effects on tissue repair, systemic inflammation, and immune function [90,91].

Though direct evidence linking HA injections to prehabilitation programs for OA remains limited, the potential role of HA injections within a comprehensive pain management strategy might be considered to improve patients’ engagement in a multicomponent physical exercise activity. In this context, the physical interventions proposed in the current literature include supervised prehabilitation programs [92] and home-based rehabilitation programs [93]. On the other hand, the sustainability of home-based exercise programs might represent a major advantage for a large-scale intervention addressing the physical deconditioning of older adults undergoing knee or hip joint replacement [94]. However, particular attention should be paid to adherence to the prehabilitation intervention, and the effective strategies addressing the potential barriers to treatment compliance are mandatory to improve outcomes in older adults with OA, including proper pain management, since pain-related fear of movement (kinesiophobia) might affect adherence to exercise interventions [95,96]. 

In addition, patient characteristics, including OA severity, prior treatment responses, and the timeline of the upcoming medical intervention, should be considered before HA injection [18,97]. Moreover, collaborative decision-making among healthcare professionals might ensure that the optimal timing and appropriateness of HA injections align with the patient’s overall prehabilitation plan. Concerns might arise from the potential increased risk of periprosthetic joint infection related to preoperative injection. In this context, the recent systematic review by Baums et al. [98] underlined that performing intra-articular corticosteroid injection within 3 months prior to total knee arthroplasty significantly increased the risk of periprosthetic joint infection, whereas no increased risk was shown when intra-articular corticosteroid injections were administered more than 6 months before surgery. 

Though a few studies assessed the role of HA injections in periprosthetic joint infection [99], the potential implications in increased infection risk should be considered before HA administration and the optimal timing for HA injection should be tailored in the comprehensive prehabilitation management of older adults with OA.

Incorporating HA injections into the prehabilitation framework might be considered as part of a holistic approach addressing multifaceted patient needs. Pain management, joint function improvement, psychological readiness, and overall well-being collectively define the patient’s prehabilitation and might be integrated into comprehensive OA management. Despite these considerations, the economic dimension of prehabilitation, including costs and cost-effectiveness, remains largely uncharted in the existing literature and the sustainability of a multicomponent prehabilitation approach is far from being fully characterized [86,87,88].

Altogether, this evidence suggests that administering HA combined with prehabilitation might provide potential advantages. The analgesic effects of HA might have a role in providing pain relief and enhancing patients’ engagement in therapeutic exercise interventions during the preoperative period. Further studies are needed to characterize the optimal timing of HA administration and the role of a multicomponent prehabilitation approach in the postoperative outcomes of patients with OA.

## 7. Limitations

To the best of our knowledge, there is a lack of literature addressing the role of visco-supplementation in the complex rehabilitation framework of osteoarthrosis. 

In this context, reviewing current evidence has highlighted a remarkable heterogeneity of participants, interventions, and outcomes, which limits the possibility of a quantitative outcomes assessment. As a consequence, our results might have been affected by the absence of standardized rehabilitation programs. Hence, only a qualitative methodological approach was pursued.

In addition, our review relied on scarce evidence in the field of robotic-assisted and technology-assisted rehabilitation programs. However, it should be noted that this limitation occurs as a consequence of the gap of knowledge in the current literature. Thus, this manuscript might represent a starting reference for paving the way of future research in the field.

## 8. Challenges and Future Perspectives

Although high-quality evidence supports multidisciplinary non-surgical approaches in the management of OA [22,23], several challenges and considerations still need to be addressed. 

In more detail, ensuring patient adherence to rehabilitation programs and self-management strategies can be challenging in older adults [100]. Factors such as motivation, understanding of the importance of adherence, and the ability to incorporate lifestyle changes can influence patient compliance [101]. Lack of adherence may hinder treatment effectiveness and limit the potential benefits of rehabilitation. In addition, the heterogeneity of OA patients in terms of age, comorbidities, disease severity, and functional limitations poses a challenge in providing individualized care [102]. A one-size-fits-all approach may not effectively address the unique needs and preferences of each patient [102].

Mobile applications and telehealth platforms can also deliver personalized exercise programs, educational materials, and interactive tools to support patients in their rehabilitation [103]. On the other hand, digital implementation might be affected by several limitations in older adults, including impediments related to patients’ hearing, vision, communication abilities, or cognitive functions [104,105]. Thus, it should be mandatory to develop personalized patient-centered strategies to effectively overcome these barriers in future research, and caregivers could assume a pivotal role in delivering optimal support to telerehabilitation [106].

In this context, the sustainability of a multidisciplinary rehabilitation program represents a hot topic in the current literature [107]. Considering the growing number of people with OA, efforts should be made toward prevention and treatments reducing the functional limitations and disabilities related to this detrimental condition, rather than focusing on giving assistance. On the other hand, resource limitations, financial constraints, and the limited availability of equipment and facilities can crucially impact the implementation of multidisciplinary care for OA. As a result, virtual reality and augmented reality systems can provide immersive rehabilitation experiences, simulating real-life movements and activities without the need for expensive equipment [75]. Mobile applications and online platforms can deliver guided exercises, educational materials, and interactive tools that can be accessed by patients using their smartphones or tablets [108]. These technologies offer scalable and affordable solutions, extending the reach of rehabilitation interventions and improving resource utilization. On the other hand, to our knowledge, no previous study assessed the effects of a telerehabilitation approach in patients undergoing intra-articular HA injection for OA.

Altogether, the previous research suggested promising features of digital implementation in the multidisciplinary management of patients with OA. Strategies such as patient education, telehealth services, resource optimization, and individualized monitoring might contribute to improved patient outcomes and enhanced patient engagement in a comprehensive rehabilitation approach including HA injection and physical therapy. 

## 9. Conclusions

In conclusion, this narrative review highlighted the potential benefits of a multidisciplinary rehabilitation of OA including hyaluronic acid intra-articular injection and exercise interventions. This comprehensive approach ensures that patients can receive holistic care tailored to their specific needs, addressing both the physical and psychosocial aspects of the OA functional impairment.

Future research should focus on the synergic effects of a comprehensive approach integrating HA injections into multidisciplinary care to enhance the knowledge of the tailored rehabilitation management of OA.

## Figures and Tables

**Figure 1 medicina-59-02047-f001:**
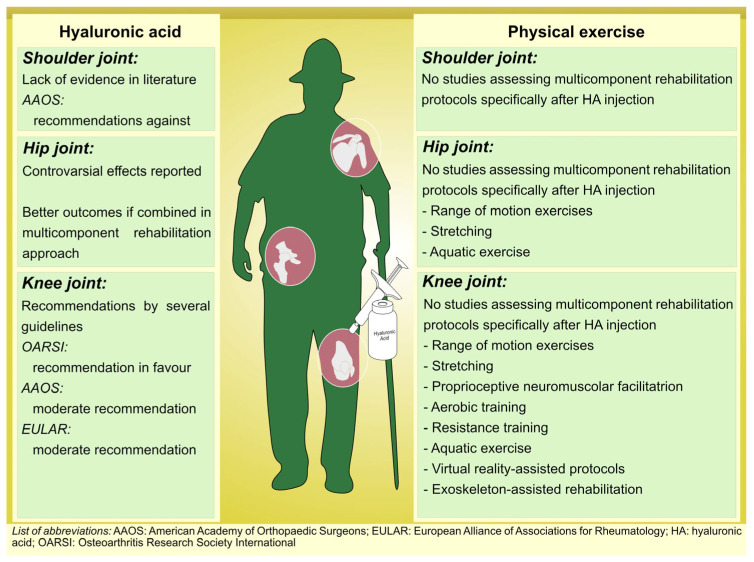
Multicomponent tailored rehabilitation protocols in patients with Osteoarthritis.

**Table 1 medicina-59-02047-t001:** Spider tool search strategy.

S	PI	D	E	R
Sample	Phenomenon of Interest	Design	Evaluation	Research Type
People with osteoarthritis treated with hyaluronic acid injection	Post-injective Rehabilitation	Any	Functional outcomes and quality of life	Qualitative
“Osteoarthritis”, “Hyaluronic Acid Injection”, “Hyaluronic Acid”, “Intra-Articular Injection”, “Viscosupplementation”	“Rehabilitation”, “Physical Therapy”, “Physical and Rehabilitation Medicine”, “Physical Exercise”, and “Physical Activity”			

**Table 2 medicina-59-02047-t002:** Search strategy.

**PubMed:** ((“osteoarthritis”[MeSH Terms] OR “osteoarthritis”[All Fields] OR “osteoarthritides”[All Fields]) AND (((“hyaluronic acid”[MeSH Terms] OR (“hyaluronic”[All Fields] AND “acid”[All Fields]) OR “hyaluronic acid”[All Fields]) AND (“inject”[All Fields] OR “injectability”[All Fields] OR “injectant”[All Fields] OR “injectants”[All Fields] OR “injectate”[All Fields] OR “injectates”[All Fields] OR “injected”[All Fields] OR “injectible”[All Fields] OR “injectibles”[All Fields] OR “injecting”[All Fields] OR “injections”[MeSH Terms] OR “injections”[All Fields] OR “injectable”[All Fields] OR “injectables”[All Fields] OR “injection”[All Fields] OR “injects”[All Fields])) AND ((“hyaluronic acid”[MeSH Terms] OR (“hyaluronic”[All Fields] AND “acid”[All Fields]) OR “hyaluronic acid”[All Fields]) AND (“inject”[All Fields] OR “injectability”[All Fields] OR “injectant”[All Fields] OR “injectants”[All Fields] OR “injectate”[All Fields] OR “injectates”[All Fields] OR “injected”[All Fields] OR “injectible”[All Fields] OR “injectibles”[All Fields] OR “injecting”[All Fields] OR “injections”[MeSH Terms] OR “injections”[All Fields] OR “injectable”[All Fields] OR “injectables”[All Fields] OR “injection”[All Fields] OR “injects”[All Fields])) OR ((“hyaluronic acid”[MeSH Terms] OR (“hyaluronic”[All Fields] AND “acid”[All Fields]) OR “hyaluronic acid”[All Fields]) AND (“inject”[All Fields] OR “injectability”[All Fields] OR “injectant”[All Fields] OR “injectants”[All Fields] OR “injectate”[All Fields] OR “injectates”[All Fields] OR “injected”[All Fields] OR “injectible”[All Fields] OR “injectibles”[All Fields] OR “injecting”[All Fields] OR “injections”[MeSH Terms] OR “injections”[All Fields] OR “injectable”[All Fields] OR “injectables”[All Fields] OR “injection”[All Fields] OR “injects”[All Fields])) OR (“injections, intra articular”[MeSH Terms] OR (“injections”[All Fields] AND “intra articular”[All Fields]) OR “intra-articular injections”[All Fields] OR (“intra”[All Fields] AND “articular”[All Fields] AND “injection”[All Fields]) OR “intra articular injection”[All Fields]) OR (“viscosupplementation”[MeSH Terms] OR “viscosupplementation”[All Fields] OR “viscosupplementations”[All Fields])) AND ((“rehabilitant”[All Fields] OR “rehabilitants”[All Fields] OR “rehabilitate”[All Fields] OR “rehabilitated”[All Fields] OR “rehabilitates”[All Fields] OR “rehabilitating”[All Fields] OR “rehabilitation”[MeSH Terms] OR “rehabilitation”[All Fields] OR “rehabilitations”[All Fields] OR “rehabilitative”[All Fields] OR “rehabilitation”[MeSH Subheading] OR “rehabilitation s”[All Fields] OR “rehabilitational”[All Fields] OR “rehabilitator”[All Fields] OR “rehabilitators”[All Fields] OR “exercise”[MeSH Terms] OR “exercise”[All Fields] OR (“physical”[All Fields] AND “exercise”[All Fields]) OR “physical exercise”[All Fields] OR “training”[All Fields] OR “train”[All Fields] OR “train s”[All Fields] OR “trained”[All Fields] OR “training s”[All Fields] OR “trainings”[All Fields] OR “trains”[All Fields])))
**Scopus:** TITLE-ABS-KEY ((Osteoarthritis) AND ((rehabilitation) OR (physical AND exercise) OR (physical AND activity)) AND ((Hyaluronic AND Acid AND Injection) OR (Hyaluronic AND Acid) OR (Intra-Articular AND Injection) OR (Viscosupplementation)))
**Web of Science:** ALL = ((Osteoarthritis) AND (rehabilitation OR physical exercise OR physical activity) AND (Hyaluronic Acid Injection OR Hyaluronic Acid OR Intra-Articular Injection OR Viscosupplementation))

## Data Availability

Not applicable.

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
