# Peer review of "Multidisciplinary Rehabilitation after Hyaluronic Acid Injections for Elderly with Knee, Hip, Shoulder, and Temporomandibular Joint Osteoarthritis"

_medicina, 2023, doi:10.3390/medicina59112047_

Round 1

Reviewer 1 Report

Comments and Suggestions for Authors

Dear Sirs,

I would like to thank you on submitting this manuscript. Although it is a well written text, I was surprised that you only approached this topic on a basic level. For example, you do not mention that the viscoelastic effect of the HA to the joint is related to its molecular weight and not all HAs have the same clinical effect. Moreover, I could not find any mention on whether the combination of a corticosteroid alters the short term or long term effect of HA. Another point that was not addressed were the biomarkers.

I am trying to point out the fact that there are an number of issues that were not discussed in this literature review. Therefore a more extensive an in-depth coverage of the literature in needed

Author Response

Reviewer 1

Dear Sirs, 

I would like to thank you on submitting this manuscript. Although it is a well written text, I was surprised that you only approached this topic on a basic level. For example, you do not mention that the viscoelastic effect of the HA to the joint is related to its molecular weight and not all HAs have the same clinical effect. Moreover, I could not find any mention on whether the combination of a corticosteroid alters the short term or long term effect of HA. Another point that was not addressed were the biomarkers. 

I am trying to point out the fact that there are an number of issues that were not discussed in this literature review. Therefore a more extensive an in-depth coverage of the literature in needed

Dear Reviewer,

We would like to thank you for your insightful comments. In accordance with your suggestions, we have now uploaded the latest version of the manuscript, providing an overview of the molecular weight of HA and its rheological properties, the potential impact of corticosteroid combinations, and the role of biomarkers, as listed below. Besides, newly added considerations and modifications have been highlighted in the attached latest version of the manuscript.  

Page 2, lines 62-70

“In this scenario, several pharmacological strategies may be used, including corticosteroid (CS), HA, or platelet-rich-plasma injections [10]. Interestingly, a recent systematic review and meta-analysis by He et al. [11] assessed the different therapeutical effects of CS injections compared to HA injections. Interestingly, CS is more effective in short-term (up to 1 month) while intriguing implications of HA intra-articular injection were underlined in the long term (up to 6 months). Thus, different combined therapies have been proposed in literature, with recent studies showing superior pain-relieving effects of combined HA and CS injections both in short-term and long-term [12,13].”.

Page 2, lines 79-98

“Moreover, HA has hydrophilic properties, enabling it to absorb water and maintain joint hydration, which is vital for optimal joint function [16] [16].  Several HA molecules are currently available in the clinical management of OA, including linear and cross-linked HA[17]. It should be noted that HA naturally occurring in healthy synovial fluid is linear, whereas cross-linked HA allows achieving higher molecular weight compounds [17]. Among the available HA formulations, current literature describes different biological effects which are related to HA molecular weight, with molecules less than 5kDa having a potential pro-inflammatory effect [18,19], and compounds having a molecular weight more than 800kDa enhancing a pro-resolving effect [19,20]. In more detail, the pro-inflammatory effect of low molecular weight HA would be driven by an up-regulatory effect of pro-inflammatory genes including nos2, tnf, il12b, and cd80, increasing the secretion of nitric oxide and TNF-α [18]. Conversely, the anti-inflammatory effect of high molecular weight HA would be driven by the up-regulation of pro-resolving genes including arg1, il10, and mrc1 [18].

In addition, a strong body of evidence describes the comprehensive therapeutic effect of high molecular weight HA, targeting both pain and function [21,22]. Apparently, these outcomes are targeted not only via the intrinsic HA physical properties, as the benefits of high molecular weight HA exceed the half-life of the compounds [20].”.

Reviewer 2 Report

Comments and Suggestions for Authors

In the manuscript: “Multidisciplinary Rehabilitation after Hyaluronic Acid Injections for Elderly with Knee, Hip, Shoulder, and Temporomandibular Joint Osteoarthritis”, the authors discussed about the multilevel benefits and synergies of integrating these two approaches in multidisciplinary OA treatment (Rehabilitation and hyaluronic acid (HA) injections).

 Overall, this manuscript results very interesting, the authors clearly explain the rational of the study and discussed the topic point by point.

However, we would like to invite the authors  to clarify some critical points:

 1.       Please check the check punctuation and spaces;

2.       The abstract is not enough clear, for example the authors talk about SANRA criteria without introducing it. Please try to adjust it;

3.       Within the introduction, the authors should introduce in general the OA treatments; with medical devices and food supplements. In this respect, the following reference should be useful; Stellavato et al. Chondroitin Sulfate in USA Dietary Supplements in Comparison to Pharma Grade Products: Analytical Fingerprint and Potential Anti-Inflammatory Effect on Human Osteoartritic Chondrocytes and Synoviocytes. Pharmaceutics. 2021 May 17;13(5):737. doi: 10.3390/pharmaceutics13050737;

4.       A briefly introduction about HA molecule may be useful;

5.       Maybe a list of abbreviations should be useful for readers;

6.       Table and figure are difficult to read, please try to adjust.

Comments on the Quality of English Language

minor mistakes of spelling are here present

Author Response

Reviewer 2

In the manuscript: “Multidisciplinary Rehabilitation after Hyaluronic Acid Injections for Elderly with Knee, Hip, Shoulder, and Temporomandibular Joint Osteoarthritis”, the authors discussed about the multilevel benefits and synergies of integrating these two approaches in multidisciplinary OA treatment (Rehabilitation and hyaluronic acid (HA) injections). 

 Overall, this manuscript results very interesting, the authors clearly explain the rational of the study and discussed the topic point by point. 

Many thanks for your letter and kind comments concerning our manuscript entitled “Multidisciplinary Rehabilitation after Hyaluronic Acid Injections for Elderly with Knee, Hip, Shoulder, and             Temporomandibular Joint Osteoarthritis”. We are glad that the Reviewer appreciated our work.

However, we would like to invite the authors  to clarify some critical points:

Newly added considerations and modifications in response have been highlighted in the attached latest version of the manuscript.

  1. Please check the check punctuation and spaces;

Thank you. A thorough full-text revision of punctuation and spacing issues was provided. (i.e., page 4, lines 148, 151).

  1. The abstract is not enough clear, for example the authors talk about SANRA criteria without introducing it. Please try to adjust it;

Thank you. The abstract has now been implemented introducing SANRA criteria (i.e., page 1, lines 24-25).

  1. Within the introduction, the authors should introduce in general the OA treatments; with medical devices and food supplements. In this respect, the following reference should be useful; Stellavato et al. Chondroitin Sulfate in USA Dietary Supplements in Comparison to Pharma Grade Products: Analytical Fingerprint and Potential Anti-Inflammatory Effect on Human Osteoartritic Chondrocytes and Synoviocytes. Pharmaceutics. 2021 May 17;13(5):737. doi: 10.3390/pharmaceutics13050737;

Thank you for your insightful suggestions. The whole section has been edited. As a result, further discussion on the importance of nutritional assessment and dietary supplement has been added.

I.e., page 2, lines 55-60.

Regarding the available treatments, growing literature is currently addressing the potential benefits of naturally derived compounds which may exert an anti-inflammatory effect and with mild or no side effects. Accordingly, oral supplements such as chondroitin-sulfate-based, and glycosamminoglycan-based products might play a role in preventive strategies and early stages of OA [8].

  1. A briefly introduction about HA molecule may be useful;

Thank you. In line with your suggestion, the introduction has now been edited.  As a result, a detailed overview of the available HA molecules was added. The rheological and biological properties of HA have been further addressed, providing a thorough description of the most up-to-date literature concerning the biomarkers changes linked to HA administration, as follows:

Page 2, lines 62-70

“In this scenario, several pharmacological strategies may be used, including corticosteroid (CS), HA, or platelet-rich-plasma injections [10]. Interestingly, a recent systematic review and meta-analysis by He et al. [11] assessed the different therapeutical effects of CS injections compared to HA injections. Interestingly, CS is more effective in short-term (up to 1 month) while intriguing implications of HA intra-articular injection were underlined in the long term (up to 6 months). Thus, different combined therapies have been proposed in literature, with recent studies showing superior pain-relieving effects of combined HA and CS injections both in short-term and long-term [12,13].”.

Page 2, lines 79-98

“Moreover, HA has hydrophilic properties, enabling it to absorb water and maintain joint hydration, which is vital for optimal joint function [16] [16].  Several HA molecules are currently available in the clinical management of OA, including linear and cross-linked HA[17]. It should be noted that HA naturally occurring in healthy synovial fluid is linear, whereas cross-linked HA allows achieving higher molecular weight compounds [17]. Among the available HA formulations, current literature describes different biological effects which are related to HA molecular weight, with molecules less than 5kDa having a potential pro-inflammatory effect [18,19], and compounds having a molecular weight more than 800kDa enhancing a pro-resolving effect [19,20]. In more detail, the pro-inflammatory effect of low molecular weight HA would be driven by an up-regulatory effect of pro-inflammatory genes including nos2, tnf, il12b, and cd80, increasing the secretion of nitric oxide and TNF-α [18]. Conversely, the anti-inflammatory effect of high molecular weight HA would be driven by the up-regulation of pro-resolving genes including arg1, il10, and mrc1 [18].

In addition, a strong body of evidence describes the comprehensive therapeutic effect of high molecular weight HA, targeting both pain and function [21,22]. Apparently, these outcomes are targeted not only via the intrinsic HA physical properties, as the benefits of high molecular weight HA exceed the half-life of the compounds [20].”.

  1. Maybe a list of abbreviations should be useful for readers;

Thank you for the suggestion. We have now added an “Abbreviations” section at the end of the manuscript (i.e., page 11, lines 501-507).

  1. Table and figure are difficult to read, please try to adjust.

Thank you. The table and figure have been reviewed in accordance with your comment, as provided on page 3, and page 7.

minor mistakes of spelling are here present

Thank you. A thorough full-text revision of typos and spelling errors was provided. As a result, the following changes were made.

Page 6, line 278: barrier for the speeding -> barrier to the speeding.

Page 8, line 333: overcome -> overcoming.